# Towards Neural Phrase-based Machine Translation

**Po-Sen Huang**[*], **Chong Wang**[*†], **Sitao Huang**[*‡], **Dengyong Zhou**[*†], **Li Deng**[*◇]
[*]Microsoft Research, [†]Google, [‡]University of Illinois at Urbana-Champaign, [◇]Citadel
`pshuang@microsoft.com`, {`chongw, dennyzhou`}`@google.com`,
`shuang91@illinois.edu`, `l.deng@ieee.org`

## Abstract

In this paper, we present Neural Phrase-based Machine Translation (NPMT).[1] Our method explicitly models the phrase structures in output sequences using Sleep-WAke Networks (SWAN), a recently proposed segmentation-based sequence modeling method. To mitigate the monotonic alignment requirement of SWAN, we introduce a new layer to perform (soft) local reordering of input sequences. Different from existing neural machine translation (NMT) approaches, NPMT does not use attention-based decoding mechanisms. Instead, it directly outputs phrases in a sequential order and can decode in linear time. Our experiments show that NPMT achieves superior performances on IWSLT 2014 German-English/English-German and IWSLT 2015 English-Vietnamese machine translation tasks compared with strong NMT baselines. We also observe that our method produces meaningful phrases in output languages.

## 1 Introduction

A word can be considered as a basic unit in languages. However, in many cases, we often need a phrase to express a concrete meaning. For example, consider understanding the following sentence, "machine learning is a field of computer science". It may become easier to comprehend if we segment it as "[machine learning] [is] [a field of] [computer science]", where the words in the bracket '[]' are regarded as "phrases". These phrases have their own meanings, and can often be reused in other contexts.

The goal of this paper is to explore the use of phrase structures aforementioned for neural network-based machine translation systems (Sutskever et al., 2014; Bahdanau et al., 2015). To this end, we develop a neural machine translation method that explicitly models phrases in target language sequences. Traditional phrase-based statistical machine translation (SMT) approaches have been shown to consistently outperform word-based ones (Koehn et al., 2003; Koehn, 2009; Lopez, 2008). However, modern neural machine translation (NMT) methods (Sutskever et al., 2014; Bahdanau et al., 2015; Luong et al., 2015) do not have an explicit treatment on phrases, but they still work surprisingly well and have been deployed to industrial systems (Zhou et al., 2016; Wu et al., 2016). The proposed Neural Phrase-based Machine Translation (NPMT) method tries to explore the advantages from both kingdoms. It builds upon Sleep-WAke Networks (SWAN), a segmentation-based sequence modeling technique described in Wang et al. (2017a), where segments (or phrases) are automatically discovered given the data. However, SWAN requires monotonic alignments between inputs and outputs. This is often not an appropriate assumption in many language pairs. To mitigate this issue, we introduce a new layer to perform (soft) local reordering on input sequences. Experimental results show that NPMT outperforms attention-based NMT baselines in terms of the BLEU score (Papineni et al., 2002) on IWSLT 2014 German-English/English-German and IWSLT 2015 English-Vietnamese translation tasks. We believe our method is one step towards the full integration of the advantages from neural machine translation and phrase-based SMT.

---

[*]Work performed while CW, DZ, and LD were at Microsoft Research and SH was interning at Microsoft Research.

[1]The source code is available at `https://github.com/posenhuang/NPMT`.

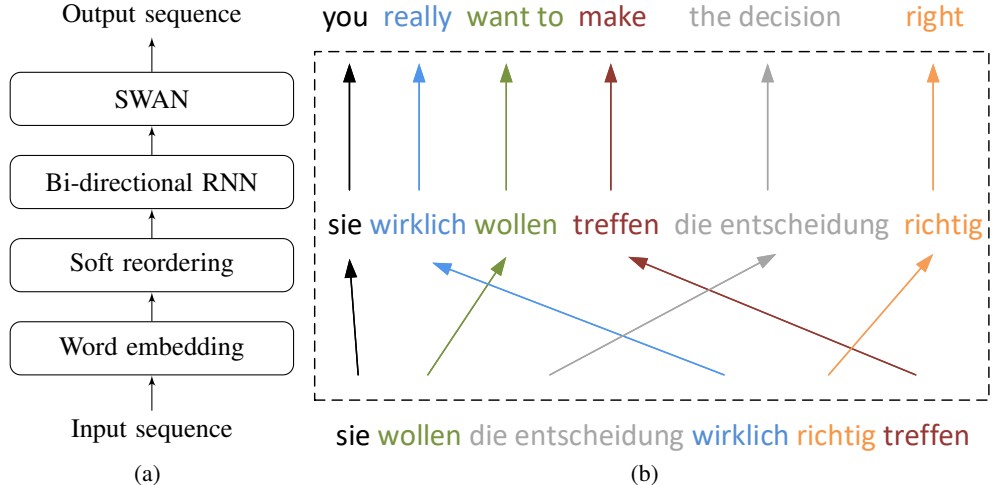

Figure 1: (a) The overall architecture of NPMT. (b) An illustration of using NPMT in German-English translation. Ideally, phrases in the source sentence (German) are first reordered. Given the new order, phrases can be translated one by one to the target phrases. These translated phrases then compose the target sentence (English). Phrase boundaries in the target language are not predefined, but automatically discovered by the model. No attention-based decoders are used here.

This paper is organized as follows. Section 2 presents the neural phrase-based machine translation model. Section 3 demonstrates the usefulness of our approach on several language pairs. We conclude our work with some discussions in Section 4.

## 2 NEURAL PHRASE-BASED MACHINE TRANSLATION

We first give an overview of the proposed NPMT architecture and some related work on incorporating phrases into NMT. We then describe the two key building blocks in NPMT: 1) SWAN, and 2) the soft reordering layer which alleviates the monotonic alignment requirement of SWAN. In the context of machine translation, we use "segment" and "phrase" interchangeably.

### 2.1 THE OVERALL ARCHITECTURE OF NPMT

Figure 1(a) shows the overall architecture of NPMT. The input sequence is first turned into embedding representations and then they go through a (soft) reordering layer (described below in Section 2.3). We then pass these "reordered" activations to the bi-directional RNN layers, which are finally fed into the SWAN layer to directly output target language in terms of segments (or phrases). While it is possible to replace bi-directional RNN layers with other layers (Gehring et al., 2017), in this paper, we have only explored this particular setting to demonstrate our proposed idea.

There have been several works that propose different ways to incorporate phrases into attention-based neural machine translation, such as Tang et al. (2016); Wang et al. (2017b); Dahlmann et al. (2017). These approaches typically use predefined phrases (obtained by external methods, e.g., phrase-based SMT) to guide or modify the existing attention-based decoder. The major difference from our approach is that, in NPMT, we do not use attention-based decoding mechanisms, and our phrase structures for the target language are automatically discovered from the training data. Another line of related work is the segment-to-segment neural transduction model (SSNT) (Yu et al., 2016), which shows promising results on a Chinese-to-English translation task under a noisy channel framework (Yu et al., 2017). In SSNT, the segments are implicit, and the monotonic alignments between the inputs and outputs are achieved using latent variables. The latent variables are marginalized out during training using dynamic programming.

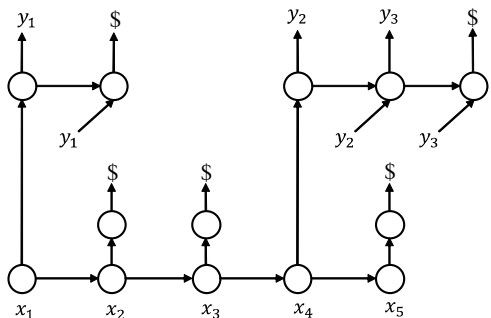

Figure 2: Courtesy to Wang et al. (2017a). Symbol $ indicates the end of a segment. Given a sequence of inputs $x_1, \ldots, x_5$, which is from the outputs from the bi-directional RNN of Figure 1(a), SWAN emits one particular segmentation of $y_{1:3} = \pi(a_{1:5})$, where $\{a_1 = \{y_1, \$\}, a_2 = \{\$\}, a_3 = \{\$\}, a_4 = \{y_2, y_3, \$\}, a_5 = \{\$\}\}$. Here $x_1$ wakes (emitting segment $a_1$) and $x_4$ wakes (emitting segment $a_4$) while $x_2$, $x_3$ and $x_5$ sleep (emitting empty segments $a_2$, $a_3$ and $a_5$ respectively).

## 2.2 Modeling phrases with SWAN

Here we review the SWAN model proposed in Wang et al. (2017a). SWAN defines a probability distribution for the output sequence given an input sequence. It models all valid output segmentations of the output sequence as well as the monotonic alignments between the output segments and the input sequence. Empty segments are allowed in the output segmentations. It does not make any assumption on the lengths of input or output sequence.

Assume input sequence for SWAN is $x_{1:T'}$, which is the outputs from bi-directional RNN of Figure 1(a), and output sequence is $y_{1:T}$. Let $\mathcal{S}_y$ denote the set containing all valid segmentations of $y_{1:T}$, with the constraint that the number of segments in a segmentation is the same as the input sequence length, $T'$. Let $a_t$ denote a segment or phrase in the target sequence. Empty segments are allowed to ensure that we can correctly align segment $a_t$ to input element $x_t$. Otherwise, we might not have a valid alignment for the input and output pair. See Figure 2 for an example of the emitted segmentation of $y_{1:T}$. The probability of the sequence $y_{1:T}$ is defined as the sum of the probabilities of all the segmentations in $\mathcal{S}_y \triangleq \{a_{1:T'} : \pi(a_{1:T'}) = y_{1:T}\}$,[2]

$$p(y_{1:T}|x_{1:T'}) \triangleq \sum_{a_{1:T'} \in \mathcal{S}_y} \prod_{t=1}^{T'} p(a_t|x_t), \qquad (1)$$

where the $p(a_t|x_t)$ is the segment probability given input element $x_t$, which is modeled using a recurrent neural network (RNN) with an additional softmax layer. $\pi(\cdot)$ is the concatenation operator and the symbol $, end of a segment, is ignored in the concatenation operator $\pi(\cdot)$. (An empty segment, which only contains $ will thus be ignored as well.) SWAN can be also understood via a generative model,

1. For $t = 1, ..., T'$:
   (a) Given an initial state of $x_t$, sample words from RNN until we reach an end of segment symbol $. This gives us a segment $a_t$.
2. Concatenate $\{a_1, ..., a_{T'}\}$ to obtain the output sequence via $\pi(a_{1:T'}) = y_{1:T}$.

Since there are more than one way to obtain the same $y_{1:T}$ using the generative process above, the probability of observing $y_{1:T}$ is obtained by summing over all possible ways, which is Eq. 1.

Note that $|\mathcal{S}_y|$ is exponentially large, direct summation quickly becomes infeasible when $T$ or $T'$ is not small. Instead, Wang et al. (2017a) developed an exact dynamic programming algorithm to tackle the computation challenges.[3] The key idea is that although the number of possible segmentations

---

[2]If predefined phrase structure information is provided for the target language in advance, we can incorporate it into SWAN by restricting the size of $\mathcal{S}_y$. We leave the exploration of this option as future work.

[3]The computational complexity of SWAN is still high even with the dynamic programming algorithm. This is the reason that it takes a longer time to train our method for larger datasets such as in WMT translation

is exponentially large, the number of possible segments is polynomial—$O(T^2)$. In other words, it is possible to first compute all possible segment probabilities, $p(a_t|x_t)$, $\forall a_t, x_t$, and then use dynamic programming to calculate the output sequence probability $p(y_{1:T}|x_{1:T'})$ in Eq. (1). The feasibility of using dynamic programming is due to a property of segmentations—a segmentation of a subsequence is also part of the segmentation of the entire sequence. In practice, a maximum length $L$ for a segment $a_t$ is enforced to reduce the computational complexity, since the length of useful segments is often not very long. Wang et al. (2017a) also discussed a way to carry over information across segments using a separate RNN, which we will not elaborate here. We refer the readers to the original paper for the algorithmic details.

SWAN defines a conditional probability for an output sequence given an input one. It can be used in many sequence-to-sequence tasks. In practice, a sequence encoder like a bi-directional RNN can be used to process the raw input sequence (like speech signals or source language) to obtain $x_{1:T'}$ that is to be passed into SWAN for decoding. For example, Wang et al. (2017a) demonstrated the usefulness of SWAN in the context of speech recognition.

Greedy decoding for SWAN is straightforward. We first note that $p(a_t|x_t)$ is modeled as an RNN with an additional softmax layer. Given each $p(a_t|x_t)$, $\forall t \in 1, \ldots, T'$, is independent of each other, we can run the RNN in parallel to produce an output segment (possibly empty) for each $p(a_t|x_t)$. We then concatenate these output segments to form the greedy decoding of the entire output sequence. The decoding satisfies the non-autoregressive property (Gu et al., 2018) and the decoding complexity is $O(T'L)$. See Wang et al. (2017a) for the algorithmic details of the beam search decoder.

We finally note that, in SWAN (thus in NPMT), only output segments are explicit; input segments are implicitly modeled by allowing empty segments in the output. This is conceptually different from the traditional phrase-based SMT where both inputs and outputs are phrases (or segments). We leave the option of exploring explicit input segments as future work.

### 2.3 LOCAL REORDERING OF INPUT SEQUENCES

SWAN assumes a monotonic alignment between the output segments and the input elements. For speech recognition experiments in Wang et al. (2017a), this is a reasonable assumption. However, for machine translation, this is usually too restrictive. In neural machine translation literature, attention mechanisms were proposed to address alignment problems (Bahdanau et al., 2015; Luong et al., 2015; Raffel et al., 2017; Vaswani et al., 2017). But it is not clear how to apply a similar attention mechanism to SWAN due to the use of segmentations for output sequences.

One may note that in NPMT, a bi-directional RNN encoder for the source language can partially mitigate the alignment issue for SWAN, since it can access every source word. However, from our empirical studies, it is not enough to obtain the best performance. Here we augment SWAN with a reordering layer that does (soft) local reordering of the input sequence. This new model leads to promising results on the IWSLT 2014 German-English/English-German, and IWSLT 2015 English-Vietnamese machine translation tasks. One additional advantage of using SWAN is that since SWAN does not use attention mechanisms, decoding can be done in parallel with linear complexity, as now we remove the need to query the entire input source for every output word (Raffel et al., 2017; Gu et al., 2018).

We now describe the details of the local reordering layer shown in Figure 3(a). Denote the input to the local reordering layer by $e_{1:T'}$, which is the output from the word embedding layer of Figure 1(a), and the output of this layer by $h_{1:T'}$, which is fed as inputs to the bi-directional RNN of Figure 1(a). We compute $h_t$ as

$$h_t = \tanh\left(\sum_{i=0}^{2\tau} \sigma\left(w_i^T[e_{t-\tau}; \ldots; e_t; \ldots; e_{t+\tau}]\right) e_{t-\tau+i}\right). \tag{2}$$

where $\sigma(\cdot)$ is the sigmoid function, and $2\tau + 1$ is the local reordering window size. Notation $[e_{t-\tau}; \ldots; e_t; \ldots; e_{t+\tau}]$ is the concatenation of vectors $e_{t-\tau}, \ldots, e_t, \ldots, e_{t+\tau}$. For $i = 0, \ldots, 2\tau$, notation $w_i$ is the parameter for the gate function at position $i$ of the input window. It decides the

---

tasks (weeks for a moderate model size). In the meantime, we are actively looking into the algorithms that can significantly speed up SWAN.

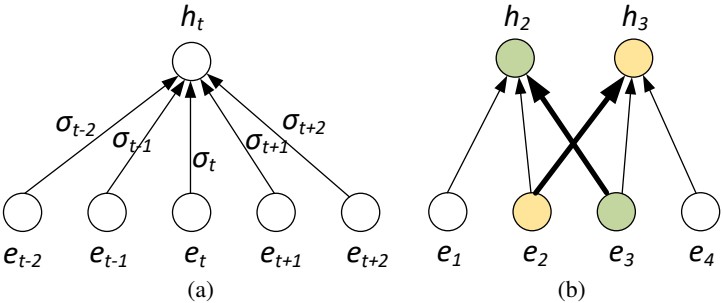

Figure 3: (a) Example of a local reordering layer of window size 5 ($\tau = 2$) to compute $h_t$. Here $\sigma_{t-2+i} \triangleq \sigma(w_i^T[e_{t-2}; e_{t-1}; e_t; e_{t+1}; e_{t+2}])$, $i = 0, \ldots, 4$, are the gates that decides how much information $h_t$ should accept from those elements from this input window. Note that all information available in this input window helps decides each gate. (b) An illustration of the reordering layer that swaps information between $e_2$ and $e_3$ and contributes to $h_3$ and $h_2$, respectively.

weight of $e_{t-\tau+i}$ through the gate $\sigma\left(w_i^T[e_{t-\tau}; \ldots; e_t; \ldots; e_{t+\tau}]\right)$. The final output $h_t$ is a weighted linear combination of the input elements, $e_{t-\tau}, \ldots, e_t, \ldots, e_{t+\tau}$, in the window followed by a non-linear transformation by the $\tanh(\cdot)$ function.

Figure 3(b) illustrates how local reordering works. Here we want to (softly) select an input element from a window given all information available in this window. Suppose we have two adjacent windows, $(e_1, e_2, e_3)$ and $(e_2, e_3, e_4)$. If $e_3$ gets the largest weight ($e_3$ is *picked*) in the first window and $e_2$ gets the largest weight ($e_2$ is *picked*) in the second window, $e_2$ and $e_3$ are effectively reordered. Our layer is different from the attention mechanism (Bahdanau et al., 2015; Luong et al., 2015; Raffel et al., 2017; Vaswani et al., 2017) in following ways. First, we do not have a query to begin with as in standard attention mechanisms. Second, unlike standard attention, which is top-down from a decoder state to encoder states, the reordering operation is bottom-up. Third, the weights $\{w_i\}_{i=0}^{2\tau}$ capture the relative positions of the input elements, whereas the weights are the same for different queries and encoder hidden states in the attention mechanism (no positional information). The reordering layer performs locally similar to a convolutional layer and the positional information is encoded by a different parameter $w_i$ for each relative position $i$ in the window. Fourth, we do not normalize the weights for the input elements $e_{t-\tau}, \ldots, e_t, \ldots, e_{t+\tau}$. This provides the reordering capability and can potentially turn off everything if needed. Finally, the gate of any position $i$ in the reordering window is determined by all input elements $e_{t-\tau}, \ldots, e_t, \ldots, e_{t+\tau}$ in the window. We provide a visualizing example of the reordering layer gates that performs input swapping in Appendix A.

One related work to our proposed reordering layer is the Gated Linear Units (GLU) (Dauphin et al., 2017) which can control the information flow of the output of a traditional convolutional layer. But GLU does not have a mechanism to decide which input element from the convolutional window to choose. From our experiments, neither GLU nor traditional convolutional layer helped our NPMT. Another related work to the window size of the reordering layer is the distortion limit in traditional phrase-based statistical machine translation methods (Brown et al., 1993). Different window sizes restrict the context of each position to different numbers of neighbors. We provide an empirical comparison of different window sizes in Appendix B.

## 3 EXPERIMENTS

In this section, we evaluate our model on the IWSLT 2014 German-English (Cettolo et al., 2014), IWSLT 2014 English-German, and IWSLT 2015 English-Vietnamese (Cettolo et al., 2015) machine translation tasks. We note that, in this paper, we limit the applications of our model to relatively small datasets to demonstrate the usefulness of our method. We plan to conduct more large scale experiments in future work.

|  | BLEU | |
|---|---|---|
|  | Greedy | Beam Search |
| MIXER (Ranzato et al., 2015) | 20.73 | 21.83 |
| LL (Wiseman & Rush, 2016) | 22.53 | 23.87 |
| BSO (Wiseman & Rush, 2016) | 23.83 | 25.48 |
| LL (Bahdanau et al., 2017) | 25.82 | 27.56 |
| LL$^*$ | 26.17 | 27.61 |
| RF-C+LL (Bahdanau et al., 2017) | 27.70 | 28.30 |
| AC+LL (Bahdanau et al., 2017) | 27.49 | 28.53 |
| NPMT (this paper) | **28.57** | **29.92** |
| NPMT+LM (this paper) | – | **30.08** |

Table 1: Translation results on the IWSLT 2014 German-English test set. MIXER Ranzato et al. (2015) uses a convolutional encoder and simpler attention. LL (attention model with log likelihood) and BSO (beam search optimization) of Wiseman & Rush (2016), and LL, RF-C+LL, and AC+LL of Bahdanau et al. (2017) use a one-layer GRU encoder and decoder with attention. (RF-C+LL and AC+LL are different settings of actor-critic algorithms combined with LL.) LL$^*$ stands for a well-tuned attention model with log likelihood with the same word embedding size, and encoder and decoder size as NPMT.

### 3.1 IWSLT14 GERMAN-ENGLISH

We evaluate our model on the German-English machine translation track of the IWSLT 2014 evaluation campaign (Cettolo et al., 2014). The data comes from translated TED talks, and the dataset contains roughly 153K training sentences, 7K development sentences, and 7K test sentences. We use the same preprocessing and dataset splits as in Ranzato et al. (2015); Wiseman & Rush (2016); Bahdanau et al. (2017). The German and English vocabulary sizes are 32,010 and 22,823 respectively.

We report our IWSLT 2014 German-English experiments using one reordering layer with window size 7, two layers of bi-directional GRU encoder (Gated recurrent unit, Chung et al. (2014)) with 256 hidden units, and two layers of unidirectional GRU decoder with 512 hidden units. We add dropout with a rate of $0.5$ in the GRU layer. We choose GRU since baselines for comparisons were using GRU. The maximum segment length is set to 6. Batch size is set as 32 (per GPU) and the Adam algorithm (Kingma & Ba, 2014) is used for optimization with an initial learning rate of 0.001. For decoding, we use greedy search and beam search with a beam size of 10. As reported in Maas et al. (2014); Bahdanau et al. (2017), we find that penalizing candidate sentences that are too short was required to obtain the best results. We add the middle term of Eq. (3) to encourage longer candidate sentences. All hyperparameters are chosen based on the development set. NPMT takes about 2–3 days to run to convergence (40 epochs) on a machine with four M40 GPUs. The results are summarized in Table 1. In addition to previous reported baselines in the literature, we also explored the best hyperparameter using the same model architecture (except the reordering layer) using sequence-to-sequence model with attention as reported as LL$^*$ of Table 1.

NPMT achieves state-of-the-art results on this dataset as far as we know. Compared to the supervised sequence-to-sequence model, LL (Bahdanau et al., 2017), NPMT achieves 2.4 BLEU gain in the greedy setting and 2.25 BLEU gain using beam-search. Our results are also better than those from the actor-critic based methods in Bahdanau et al. (2017). But we note that our proposed method is orthogonal to the actor-critic method. So it is possible to further improve our results using the actor-critic method.

We also run the following two experiments to verify the sources of the gain. The first is to add a reordering layer to the original sequence-to-sequence model with attention, which gives us BLEU scores of 25.55 (greedy) and 26.91 (beam search). Since the attention mechanism and reordering layer capture similar information, adding the reordering layer to the sequence-to-sequence model with attention does not improve the performance. The second is to remove the reordering layer from NPMT, which gives us BLEU scores of 27.79 (greedy) and 29.28 (beam search). This shows that the reordering layer and SWAN are both important for the effectiveness of NPMT.

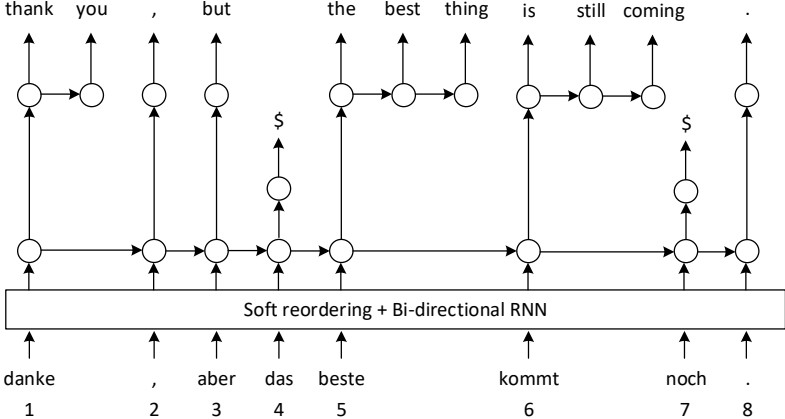

Figure 4: An example of NPMT greedy decoding output for German-English translation. The example corresponds to the first example of Table 2. Note that for illustrating the input and output segments, we do not take into account of the behavior of the reordering layer and bi-directional RNN—the index mappings from source to target assumes monotonic alignments so some of them might be inaccurate.

| | |
|---|---|
| source | $^1$danke $^2$, $^3$aber $^4$das $^5$beste $^6$kommt $^7$noch $^8$. |
| greedy decoding | $^1$thank you • $^2$, • $^3$but • $^5$the best thing • $^6$is still coming • $^8$. |
| target ground truth | thanks . i haven 't come to the best part . |
| source | $^1$sie $^2$können $^3$einen $^4$schalter $^5$dazwischen $^6$einfügen $^7$und $^8$so $^9$haben $^{10}$sie $^{11}$einen $^{12}$kleinen $^{13}$UNK $^{14}$erstellt $^{15}$. |
| greedy decoding | $^1$you can put • $^4$a switch • $^5$in between • $^7$and • $^8$so • $^{10}$they created • $^{12}$a little • $^{13}$UNK $^{14}$. |
| target ground truth | you can put a knob in between and now you 've made a little UNK . |
| source | $^1$sie $^2$wollen $^3$die $^4$entscheidung $^5$wirklich $^6$richtig $^7$treffen $^8$, $^9$wenn $^{10}$es $^{11}$für $^{12}$alle $^{13}$ewigkeit $^{14}$ist $^{15}$, $^{16}$richtig $^{17}$? |
| greedy decoding | $^1$you really want to make • $^4$the decision • $^6$right • $^8$, • $^9$if • $^{10}$it 's • $^{11}$for • $^{12}$all • $^{13}$eternity • $^{15}$, • $^{16}$right • $^{17}$? |
| target ground truth | you really want to get the decision right if it 's for all eternity , right ? |
| source | $^1$es $^2$gibt $^3$zehntausende $^4$maschinen $^5$rund $^6$um $^7$die $^8$welt $^9$die $^{10}$kleine $^{11}$stücke $^{12}$von $^{13}$dna $^{14}$herstellen $^{15}$können $^{16}$, $^{17}$30 $^{18}$bis $^{19}$50 $^{20}$buchstaben $^{21}$lang $^{22}$aber $^{23}$es $^{24}$ist $^{25}$ein $^{26}$UNK $^{27}$prozess $^{28}$, $^{29}$also $^{30}$je $^{31}$länger $^{32}$man $^{33}$ein $^{34}$stück $^{35}$macht $^{36}$, $^{37}$umso $^{38}$mehr $^{39}$fehler $^{40}$passieren $^{41}$. |
| greedy decoding | $^1$there are • $^3$tens of thousands of • $^4$machines • $^6$around • $^8$the world • $^9$can make • $^{10}$little • $^{11}$pieces • $^{12}$of • $^{13}$dna • $^{16}$, • $^{17}$30 • $^{18}$to • $^{19}$50 • $^{20}$letters • $^{21}$long • $^{22}$, but • $^{23}$it 's • $^{26}$a more UNK • $^{27}$process • $^{28}$, • $^{29}$so • $^{31}$the longer • $^{32}$you make • $^{34}$a piece • $^{36}$, • $^{38}$the more • $^{39}$mistakes • $^{40}$happen • $^{41}$. |
| target ground truth | there are tens of thousands of machines around the world that make small pieces of dna – 30 to 50 letters - in length - and it 's a UNK process , so the longer you make the piece , the more errors there are . |

Table 2: Examples of German-English translation outputs with their segmentations. We label the indexes of the words in the source sentence and we use those indexes to indicate where the output segment is emitted. For example, in greedy decoding results, "$^i$word$_1$,...,word$_m$" denotes $i$-th word in the source sentence emits words word$_1$,...,word$_m$ during decoding (assuming monotonic alignments). The "•" represents the segment boundary in the target output. See Figure 4 for a visualization of row 1 in this table.

In greedy decoding, we can estimate the *average segment length*[4] for the output. The average segment length is around 1.4–1.6, indicating phrases with more than one word are being decoded. Figure 4 shows an example of the input and decoding results with NPMT. We can observe phrase-level translation being captured by the model (e.g., "danke" → "thank you"). The model also knows when to *sleep* before outputting a phrase (e.g., "das" → "$"). We use the indexes of words in the source sentence to indicate where the output phrases are from. Table 2 shows some sampled exam-

---

[4]The average segment length is defined as the length of the output (excluding end of segment symbol $) divided by the number of segments (not counting the ones only containing $).

ples. We can observe there are many informative segments in the decoding results, e.g., "tens of thousands of", "the best thing", "a little", etc. There are also mappings from phrase to phrase, word to phrases, and phrase to word in the examples. Following the analysis, we show the most frequent phrase mappings in Appendix C.

We also explore an option of adding a language-model score during beam search as the traditional statistical machine translation does. This option might not make much sense in attention-based approaches, since the decoder itself is usually a neural network language model. In SWAN, however, there is no language models directly involved in the segmentation modeling,[5] and we find it useful to have an external language model during beam search. We use a 4th-order language model trained using the KenLM implementation (Heafield et al., 2013) for English target training data. So the final beam search score we use is

$$Q(y) = \log p(y|x) + \lambda_1 \text{word\_count}(y) + \lambda_2 \log p_{\text{lm}}(y), \tag{3}$$

where we empirically find that $\lambda_1 = 1.2$ and $\lambda_2 = 0.2$ give good performance, which are tuned on the development set. The results with the external language model are denoted by NPMT+LM in Table 1. If no external language models are used, we set $\lambda_2 = 0$. This scoring function is similar to the one for speech recognition in Hannun et al. (2014).

## 3.2 IWSLT14 English-German

We also evaluate our model on the opposition direction, English-German, which translates from a more segmented text to a more inflectional one. Following the setup in Section 3.1, we use the same dataset with the opposite source and target languages. We use the same model architecture, optimization algorithm and beam search size as the German-English translation task. NPMT takes about 2–3 days to run to convergence (40 epochs) on a machine with four M40 GPUs.

Given there is no previous sequence-to-sequence attention model baseline for this setup, we create a strong one and tune hyperparameters on the development set. The results are shown in Table 3. Based on the development set, we set $\lambda_1 = 1$ and $\lambda_2 = 0.15$ in Eq. (3). Our model outperforms sequence-to-sequence model with attention by 2.46 BLEU and 2.49 BLEU in greedy and beam search cases. We can also use a 4th-order language model trained using the KenLM implementation for German target training data, which further improves the performance. Some sampled examples are shown in Table 4. Several informative segments/phrases can be found in the decoding results, e.g., "some time ago" → "vor enniger zeit".

|  | BLEU | |
| --- | --- | --- |
|  | Greedy | Beam Search |
| Sequence-to-sequence with attention | 21.26 | 22.59 |
| NPMT (this paper) | **23.62** | **25.08** |
| NPMT+LM (this paper) | – | **25.36** |

Table 3: Translation results on the IWSLT 2014 English-German test set.

## 3.3 IWSLT15 English-Vietnamese

In this section, we evaluate our model on the IWSLT 2015 English to Vietnamese machine translation task. The data is from translated TED talks, and the dataset contains roughly 133K training sentence pairs provided by the IWSLT 2015 Evaluation Campaign (Cettolo et al., 2015). Following the same preprocessing steps in Luong & Manning (2015); Raffel et al. (2017), we use the TED tst2012 (1553 sentences) as a validation set for hyperparameter tuning and TED tst2013 (1268 sentences) as a test set. The Vietnamese and English vocabulary sizes are 7,709 and 17,191 respectively.

---

[5]In Wang et al. (2017a), SWAN does have an option to use a separate RNN that connects the segments, which can be seen as a language model. However, different from speech recognition experiments, we find in machine translation experiments, adding this separate RNN leads to a worse performance. We suspect this is because an RNN language model can be easier to learn than the segmentation structures and SWAN gets stuck in that local mode. This is further evidenced by the fact that the average segment length is much shorter with a separate RNN in SWAN.

| | |
|---|---|
| source | [1]how [2]would [3]you [4]guys [5]describe [6]your [7]brand [8]? |
| greedy decoding | [1]wie • [2]würdet • [3]sie • [6]ihre marke • [8]beschreiben ? |
| target ground truth | wie würdet ihr eure marke beschreiben ? |
| source | [1]if [2]the [3]museum [4]has [5]given [6]us [7]the [8]image [9], [10]you [11]click [12]on [13]it [14]. |
| greedy decoding | [1]wenn • [2]das museum • [6]uns • [7]das bild • [9]gegeben hat ,• [10]klicken sie • [13]darauf • [14]. |
| target ground truth | wenn das museum uns das bild gegeben hat , klicken sie darauf . |
| source | [1]they [2]are [3]frustrated [4]as [5]hell [6]with [7]it [8], [9]but [10]they [11]'re [12]not [13]complaining [14]about [15]it [16], [17]they [18]'re [19]fixing [20]it [21]. |
| greedy decoding | [1]sie sind • [3]frustriert • [8], • [9]aber • [10]sie UNK sich • [12]nicht • [15]darüber • [16], • [17]sie reparieren • [20]es • [21]. |
| target ground truth | sie sie sind fürchterlich frustriert mit ihr , aber sie beschweren sich nicht darüber , sie reparieren sie . ? |
| source | [1]now [2]some [3]time [4]ago [5], [6]if [7]you [8]wanted [9]to [10]win [11]a [12]formula [13]1 [14]race [15], [16]you [17]take [18]a [19]budget [20], [21]and [22]you [23]bet [24]your [25]budget [26]on [27]a [28]good [29]driver [30]and [31]a [32]good [33]car [34]. |
| greedy decoding | [2]vor einiger zeit • [6]wenn • [7]man • [11]eine formel • [15]gewinnen will , • [18]ein budget • [21]und • [23], dass • [24]ihr budget • [27]auf einem guten • [29]fahrer • [30]und • [31]ein gutes • [33]auto • [34]. |
| target ground truth | vor einiger zeit war es so , dass wenn sie ein formel 1 rennen gewinnen wollten , dann nahmen sie ihr budget und setzten ihr geld auf einen guten fahrer und ein gutes auto . |

Table 4: Examples of English-German translation outputs with their segmentations. The meanings of the superscript indexes and the "•" symbol are the same as those in Table 2.

We use one reordering layer with window size 7, two layers of bi-directional LSTM (Long short-term memory, Hochreiter & Schmidhuber (1997)) encoder with 512 hidden units, and three layers of unidirectional LSTM decoder with 512 hidden units. We add dropout with a rate of $0.4$ in the LSTM layer. We choose LSTM since baselines for comparisons were using LSTM. The maximum segment length is set to 7. Batch size is set as 48 (per GPU) and the Adam algorithm Kingma & Ba (2014) is used for optimization with an initial learning rate of 0.001. For decoding, we use greedy decoding and beam search with a beam size of 10. The results are shown in Table 5. Based on the development set, we set $\lambda_1 = 0.7$ and $\lambda_2 = 0.15$ in Eq. (3). NPMT takes about one day to run to convergence (15 epochs) on a machine with 4 M40 GPUs. Our model outperforms sequence-to-sequence model with attention by 1.41 BLEU and 1.59 BLEU in greedy and beam search cases. We also use a 4th-order language model trained using the KenLM implementation for Vietnamese target training data, which further improves the BLEU score. Note that our reordering layer relaxes the monotonic assumption as in Raffel et al. (2017) and is able to decode in linear time. Empirically we outperform models with monotonic attention. Table 6 shows some sampled examples.

| | BLEU | |
|---|---|---|
| | Greedy | Beam Search |
| Hard monotonic (Raffel et al., 2017) | 23.00 | - |
| Luong & Manning (2015) | - | 23.30 |
| Sequence-to-sequence model with attention | 25.50 | 26.10 |
| NPMT (this paper) | **26.91** | **27.69** |
| NPMT+LM (this paper) | – | **28.07** |

Table 5: Translation results on the IWSLT 2015 English-Vietnamese tst2013 test set. The result of the sequence-to-sequence model with attention is obtained from an open source model provided by the authors.[7]

## 4 CONCLUSION

We proposed NPMT, a neural phrase-based machine translation system that models phrase structures in the target language using SWAN. We also introduced a local reordering layer to mitigate the

---

[7]https://github.com/tensorflow/nmt

| | |
|---|---|
| source | [1]And [2]I [3]figured [4], [5]this [6]has [7]to [8]stop [9]. |
| greedy decoding | [1]Và ● [2]tôi ● [3]nhận ra rằng ● [4], ● [5]điều này ● [6]phải● [8]dừng lại [9]. |
| target ground truth | Và tôi nhận ra rằng điều đó phải chấm dứt . |
| source | [1]So [2]great [3]progress [4]and [5]treatment [6]has [7]been [8]made [9]over [10]the [11]years [12]. |
| greedy decoding | [1]Vì vậy , ● [2]tiến bộ● [4]và ● [5]điều trị ● [6]đã ● [7]được ● [8]tạo ra ● [9]trong ● [10]những ● [11]năm ● [12]. |
| target ground truth | Trong suốt những năm qua đã có sự  tiến bộ to lớn trong quá trình điều trị . |
| source | [1]The [2]passion [3]that [4]the [5]person [6]has [7]for [8]her [9]own [10]growth [11]is [12]the [13]most [14]important [15]thing [16]. |
| greedy decoding | [1]Niềm đam mê ● [3]rằng ● [5]người ● [6]có ● [7]cho ● [8]sự phát triển ● [10]của cô ấy ● [11]là ● [13]điều ● [14]quan trọng ● [15]nhất ● [16]. |
| target ground truth | Cái khát vọng của người phụ nữ có cho sự phát triển của bản thân là thứ quan trọng nhất . |
| source | [1]We [2]have [3]eight [4]species [5]of [6]UNK [7]that [8]occur [9]in [10]Kenya [11], [12]of [13]which [14]six [15]are [16]highly [17]threatened [18]with [19]extinction [20]. |
| greedy decoding | [1]Chúng ta ● [2]có ● [3]8 ● [4]loài ● [6]UNK ● [8]xảy ra ● [9]ở ● [10]Kenya ● [11], ● [14]6 ● [17]bị đe doạ ● [19]tuyệt chủng ● [20]. |
| target ground truth | Chúng ta có 8 loài kền kền xuất hiện tại Kenya , trong đó có 6 loài bị đe doạ với nguy cơ  tuyệt chủng cao . |

Table 6: Examples of English-Vietnamese translation outputs with their segmentations. The meanings of the superscript indexes and the "●" symbol are the same as those in Table 2.

monotonic alignment requirement in SWAN. Our experimental results showed promising results on IWSLT 2014 German-English, English-German, and IWSLT 2015 English-Vietnamese machine translation tasks. The results suggest that NPMT can potentially be extended to explore the structures in other challenging sequence-to-sequence problems. In future work, we will explore two directions: 1) speed up NPMT and apply it to larger datasets and more language pairs; 2) investigate how to learn input and output phrases simultaneously.

## 5 ACKNOWLEDGMENTS

We thank Jacob Devlin, Adith Swaminathan, Frank Seide, Xiaodong He, and anonymous reviewers for their valuable feedback.

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

## A    REORDERING LAYER ANALYSIS

To further understand the behavior of the reordering layer, we examine the values of the gate $\sigma\left(w_i^T[e_{t-\tau}; \ldots; e_t; \ldots; e_{t+\tau}]\right)$ in Eq. (2). We study the NPMT English-German model in Section 3.2. In Figure 5, we show an example that translates from "can you translate it ?" to "können man es übersetzen ?", where the mapping between words are as follows: "can → können", "you → man", "translate → übersetzen", "it → es" and "? → ?". Note that the example needs to be reordered from "translate it" to "es übersetzen". Each row of Figure 5 represents a window of size 7 that is centered at a source sentence word. The values in the matrix represent the gate values for the corresponding words. The gate values will later be multiplied with the embedding $e_{t-\tau+i}$ of Eq. (2) and contribute to the hidden vector $h_t$. The y-axis represents the word/phrases emitted from the corresponding position. We can observe that the gates mostly focus on the central word since the first part of the sentence only requires monotonic alignment. Interestingly, the model outputs "$" (empty) when the model has the word "translate" in the center of the window. Then, the model outputs "es" when the model encounters "it". Finally, in the last window (top row), the model not only has a large gate value to the center input "?", but the model also has a relatively large gate value to the word "translate" in order to output the translation "übersetzen ?". This shows an example of the reordering effect achieved by using the gating mechanism of the reordering layer.

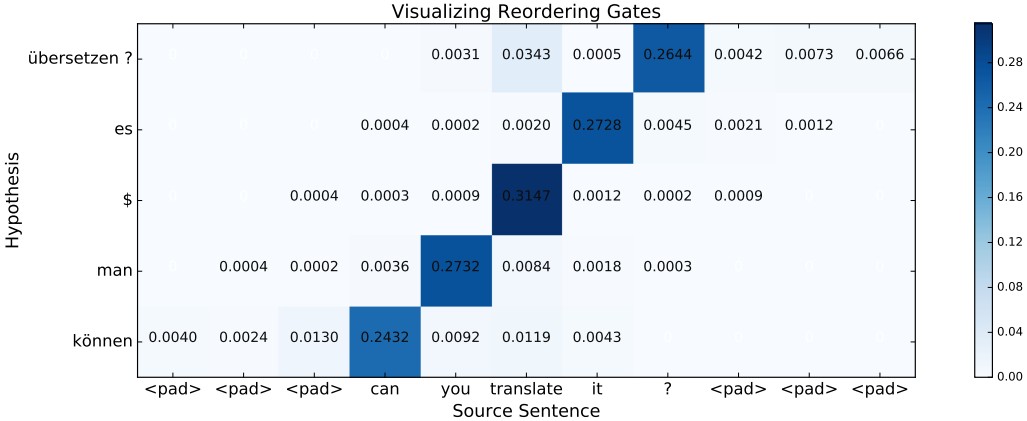

Figure 5: Visualizing reordering gates in the NPMT English-German translation model.

## B    EFFECT OF WINDOW SIZES IN THE REORDERING LAYER

In this section, we examine the effect of window sizes in the reordering layer. Following the setup in Section 3.2, we evaluate the performance of different window sizes on the IWSLT 2014 English-German translation task. Table 7 summarizes the results. We can observe that the performance reaches the peak with a windows size of 7. With a window size of 5, the performance drops 0.88 BLEU in greedy decoding and 0.72 BLEU using beam search. It suggests that the context window is not large enough to properly perform reordering. When the window sizes are 9 and 11, we do not observe further improvements. It might be because the translation between English and German mostly requires local word reordering.

## C    PHRASE MAPPING EXAMPLES

Following the examples of Table 2, we analyze the decoding results on the test set of the German-English translation task. Given we do not have explicit input segments in NPMT, we assume input words that emit "$" symbol are within the same group as the next non-'$' word. For example, in Figure 4, input words "das beste" are considered as an input segment. We then can aggregate all the input, output segments (phrases) and sort them based on the frequency. Tables C and C show the most-frequent input, output phrase mappings.

| Window Size | BLEU | |
| --- | --- | --- |
| | Greedy | Beam Search |
| 5 | 22.74 | 24.36 |
| 7 | **23.62** | **25.08** |
| 9 | 23.11 | 24.68 |
| 11 | 23.12 | 24.65 |

Table 7: Analyze the effect of reordering layer window sizes in translation results on the IWSLT 2014 English-German test set.

| One → One | One → Many | Many → One | Many → Many | Many → Many* |
| --- | --- | --- | --- | --- |
| , → , | es → it 's | , dass → that | die UNK → the UNK | wissen sie → you know |
| . → . | UNK → the UNK | in der → in | der UNK → the UNK | in diesem → in this |
| und → and | und → , and | UNK . → . | ein UNK → a UNK | die welt → the world |
| UNK → UNK | das → this is | UNK , → , | das UNK → the UNK | ist es → it 's |
| aber → but | das, → that 's | , die → that | eine UNK → a UNK | " . → . " |
| " → " | UNK → a UNK | ist . → . | in UNK → in UNK | ein paar → a few |
| ist → is | ich → i think | in den → in | den UNK → the UNK | gibt es → there 's |
| der → of | es → it was | ist , → , | wissen sie → you know | der welt → the world |
| von → of | dies → this is | sind . → . | in diesem → in this | die frage → the question |
| mit → with | es → there 's | , wenn → if | dem UNK → the UNK | haben wir → we have |

Table 8: German-English phrase mapping results. We show the top 10 input, output phrase mappings in five categories ("One" stands for single word and "Many" stands for multiple words.). In the last column, Many → Many*, we remove the phrases with the "UNK" word as the "UNK" appears often.

| Phrases with 3 words | Phrases with 4 words |
| --- | --- |
| auf der ganzen → all over the | auf der ganzen → a little bit of |
| gibt eine menge → a lot of | weiß nicht , was → what 's going to be |
| dann hat er→ he doesn 't have | tun , das wir → we can 't do |
| , die man → you can do | tat , das ich → i didn 't do |
| das können wir → we can do that | zu verbessern , die → that can be done |

Table 9: German-English longer phrase mapping results. We show the top 5 input, output phrase mappings for two categories: input and output phrases with three words, and input and output phrases with four words.

