# OpenReview forum: "Towards Neural Phrase-based Machine Translation"
_ICLR.cc/2018/Conference — Accept (Poster)_

### Official Review · AnonReviewer2 · 2017-11-23

**Rating:** 6
**Confidence:** 3

**Review:**

The paper introduces a neural translation model that automatically discovers phrases.  This idea is very interesting and tries to marry phrase-based statistical machine translation with neural methods in a principled way. However, the clarity of the paper could be improved.

The local reordering layer has the ability to swap inputs, however, how do you ensure that it actually does swap inputs rather than ignoring some inputs and duplicating others?

Are all segments translated independently, or do you carry over the hidden state of the decoder RNN between segments? In Figure 1 both a BRNN and SWAN layer are shown, is there another RNN in the SWAN layer, or does the BRNN emit the final outputs after the segments have been determined?

---

> ### Author Response · Authors · 2017-12-17
> **Responses to Reviewer2**
>
> Thank you for your valuable comments.  We address the comments and questions below:
> 1. The local reordering layer has the ability to swap inputs, however, how do you ensure that it actually does swap inputs rather than ignoring some inputs and duplicating others?
> <Response>: We do not have a guarantee that the layer forces to swap inputs as it is data driven. In Appendix A, we show an example translating from "can you translate it ?" to  "können es übersetzen?" to show that some input information is swapped.  Note that the example needs to be reordered from "translate it" to "es übersetzen".  Each row of Figure 3 represents a window of size 7 that is centered at a source sentence word. We can observe that the gates mostly focus on the central word since the first part of the sentence only requires monotonic alignment. Interestingly, the model outputs "$" (empty) when the model has the word "translate" in the center of the window.  Then, the model outputs "es" when the model encounters "it".  Finally, in the last window (top row), the model not only has a large gate value to the center input "?", but the model also has a relatively large gate value to the word "translate" in order to output the translation "übersetzen ?".  This shows an example of the reordering effect achieved by using the gating mechanism of the reordering layer.
>
> 2. Are all segments translated independently, or do you carry over the hidden state of the decoder RNN between segments?
> <Response>: Yes, all the segments are translated independently. We do not carry over the hidden states between segments. Hence, the decoding can be parallelized. We are highlighting this part in the second to last paragraph of Section 2.2.
>
> 3. In Figure 1 both a BRNN and a SWAN layer are shown, is there another RNN in the SWAN layer, or does the BRNN emit the final outputs after the segments have been determined?
> <Response>: In Figure 1, the reordering layer and BRNN can be considered as the encoder of an input sequence. The SWAN is the decoder, which contains another unidirectional RNN for p(a_t|x_t) in Eq. (1). The BRNN emits x_t to SWAN. We added some clarification in defining x_t to address it.

---

### Official Review · AnonReviewer3 · 2017-11-27
**Reasonable modeling but some unclear point**

**Rating:** 6
**Confidence:** 4

**Review:**

Authors proposed a new neural-network based machine translation method that generates the target sentence by generating multiple partial segments in the target sentence from different positions in the source information. The model is based on the SWAN architecture which is previously proposed, and an additional "local reordering" layer to reshuffle source information to adjust those positions to the target sentence.

Using the SWAN architecture looks more reasonable than the conventional attention mechanism when the ground-truth word alignment is monotone. Also, the concept of local reordering mechanism looks well to improve the basic SWAN model to reconfigure it to the situation of machine translation tasks.

The "window size" of the local reordering layer looks like the "distortion limit" used in traditional phrase-based statistical machine translation methods, and this hyperparameter may impose a similar issue with that of the distortion limit into the proposed model; small window sizes may drop information about long dependency. For example, verbs in German sentences sometimes move to the tail of the sentence and they introduce a dependency between some distant words in the sentence. Since reordering windows restrict the context of each position to a limited number of neighbors, it may not capture distant information enough. I expected that some observations about this point will be unveiled in the paper, but unfortunately, the paper described only a few BLEU scores with different window sizes which have not enough information about it. It is useful for all followers of this paper to provide some observations about this point.
In addition, it could be very meaningful to provide some experimental results on linguistically distant language pairs, such as Japanese and English, or simply reversing word orders in either source or target sentences (this might work to simulate the case of distant reordering).

Authors argued some differences between conventional attention mechanism and the local reordering mechanism, but it is somewhat unclear that which ones are the definite difference between those approaches.

A super interesting and mysterious point of the proposed method is that it achieves better BLEU than conventional methods despite no any global language models (Table 1 row 8), and the language model options (Table 1 row 9 and footnote 4) may reduce the model accuracy as well as it works not so effectively. This phenomenon definitely goes against the intuitions about developing most of the conventional machine translation models. Specifically, it is unclear how the model correctly treats word connections between segments without any global language model. Authors should pay attention to explain more detailed analysis about this point in the paper.

Eq. (1) is incorrect. According to Fig. 2, the conditional probability in the product operator should be revised to p(a_t | x_{1:t}, a_{1:t-1}), and the independence approximation to remove a_{1:t-1} from the conditions should also be noted in the paper.
Nevertheless, the condition x_{1:t} could not be reduced because the source position is always conditioned by all previous positions through an RNN.

---

> ### Author Response · Authors · 2017-12-17
> **Responses to Reviewer3**
>
> Thank you for your valuable comments.  We address the comments and questions below:
> 1. The "window size" of the local reordering layer looks like the "distortion limit" used in traditional phrase-based statistical machine translation methods, and this hyperparameter may impose a similar issue with that of the distortion limit into the proposed model.
> <Response>: Thanks a lot for your suggestion. We add the reference [Brown 1993] and discussion to the end of Section 2.3. We believe the limit of local reordering is mitigated by using bidirectional RNN after that. Thus it is not very clear how to analyze the exact behavior of the local reordering layer. We are currently actively investigating new ways of doing so.
>
> 2. It could be very meaningful to provide some experimental results on linguistically distant language pairs, such as Japanese and English, or simply reversing word orders in either source or target sentences.
> <Response>: Thanks a lot for your suggestion. This is definitely one important direction we should investigate in future work.
>
> 3. Authors argued some differences between conventional attention mechanism and the local reordering mechanism, but it is somewhat unclear that which ones are the definite difference between those approaches.
> <Response>: We reiterate and reorganize the important differences here:
> First, we do not have a query to begin with as in standard attention mechanisms. Second, unlike standard attention, which is top-down from a decoder state to encoder states, the reordering operation is bottom-up. Third, the weights {w_i}_{i=0}^{2\tau} capture the relative positions of the input elements, whereas the weights are the same for different queries and encoder hidden states in the attention mechanism (no positional information). The reordering layer performs locally similar to a convolutional layer and the positional information is encoded by a different parameter w_i for each relative position i in the window. Fourth, we do not normalize the weights for the input elements e_{t-\tau}, ..., e_t, ..., e_{t+\tau}.  This provides the reordering capability and can potentially turn off everything if needed. Finally, the gate of any position i in the reordering window is determined by all input elements e_{t-\tau},…, e_t, …, e_{t+\tau} in the window.
>
> 4. Equation (1) is incorrect. According to Fig. 2, the conditional probability in the product operator should be revised to p(a_t | x_{1:t}, a_{1:t-1}), and the independence approximation to remove a_{1:t-1} from the conditions should also be noted in the paper. Nevertheless, the condition x_{1:t} could not be reduced because the source position is always conditioned by all previous positions through an RNN.
> <Response>: We respectfully disagree with this assessment. The Eq. (1) is not an approximation; it is the way we model the output. This is motivated by Eqs. (2) and (3) of the CTC paper [Graves 2006],  p(y_{1:T}|x_{1:T’}) = sum_{a_{1:T’}} p(a_{1:T}|x_{1:T’}) marginalizes over the set of all possible segmentations. And a_{1:T’} is a collection of the segments that, when concatenated, leads to y_{1:T}. We also have p(a_{1:T}|x_{1:T’}) = \prod{t=1}^{T’} p(a_t|x_t) given the assumption that the outputs at different times are conditional independent given the input state x_t. Put it in another way, our approach can be described via a fully generative model:
>
> For t=1, T’:
> 	Using x_t as the initial state, sample target words from RNN until we reach the end of segment symbol. This gives us segment a_t.
> Finally, concatenate {a_1, ...a_T’} to obtain an output y_{1:T}.
>
> Since there are more than one way to obtain the same y_{1:T}, its probability becomes p(y_{1:T}|x_{1:T’}) = sum_{a_{1:T’}, where a_{1:T’}\in S_y} p(a_{1:T}|x_{1:T’}), the Eq. (1) in our paper. This explanation is also added in the updated paper.
>
> [Graves 2006] Graves, Alex, et al. "Connectionist temporal classification: labeling unsegmented sequence data with recurrent neural networks." Proceedings of the 23rd international conference on Machine learning. ACM, 2006.
>
> 5. NPMT achieves better BLEU than conventional methods despite no any global language models (Table 1 row 8), and the language model options (Table 1 row 9 and footnote 4) may reduce the model accuracy as well as it works not so effectively.
> <Response>: We also believe this is a super interesting and exciting observation. Our current understanding is that phrases are important building blocks of the whole target sentence and these phrases are relatively independent. With the help of being able to see the entire input sentence through the encoder, the performance can still be quite good without modeling the connection between the phrases. This is exciting because decoding can be done in linear time and can also be parallelized. We also show that adding an n-gram LM during beam search did help improve performance (Table 1 row 9).

---

### Official Review · AnonReviewer1 · 2017-11-29
**This paper introduces a new architecture for end to end neural machine translation with promising results on small datasets.**

**Rating:** 8
**Confidence:** 5

**Review:**

This paper introduces a new architecture for end to end neural machine translation. Inspired by the phrase based approach, the translation process is decomposed as follows : source words are embedded and then reordered; a bilstm then encodes the reordered source; a sleep wake network finally generates the target sequence as a phrase sequence built from left to right.

This kind of approach is more related to ngram based machine translation than conventional phrase based one.

The idea is nice. The proposed approach does not rely on attention based model. This opens nice perpectives for better and faster inference.

My first concern is about the architecture description. For instance, the swan part is not really stand alone. For reader who does not already know this net, I'm not sure this is really clear. Moreover, there is no link between notations used for the swan part and the ones used in the reordering part.

Then, one question arises. Why don't you consider the reordering of the whole source sentence. Maybe you could motivate your choice at this point. This is the main contribution of the paper, since swan already exists.

Finally, the experimental part shows nice improvements but: 1/ you must provide baseline results with a well tuned phrase based mt system; 2/ the datasets are small ones, as well as the vocabularies, you should try with larger datasets and bpe for sake of comparison.

---

> ### Author Response · Authors · 2017-12-17
> **Responses to Reviewer1**
>
> Thank you for your valuable comments.  We address the comments and questions below:
> 1. For instance, the swan part is not really stand alone. For reader who does not already know this net, I'm not sure this is really clear.
> <Response>: We add some detailed explanations to address the confusion from Reviewer2 (see responses to Reviewer2) and make SWAN more stand alone.  For example, we added an explanation of SWAN via a probabilistic generative model in section 2.2.
>
> 2. There is no link between notations used for the swan part and the ones used in the reordering part
> <Response>: We clarify the description of symbols h and x, and indicate the connections between the two using the bi-directional RNN in Figure 1 (a).
>
> 3. Why don't you consider the reordering of the whole source sentence?
> <Response>: Our proposed reordering layer is not limited to local reordering. Empirically, we found the results do not improve when we increase the window sizes in our experiments, see Appendix B for details. It might be due to the choice of language pairs, which are relative monotonic.
>
> 4. The experimental part shows nice improvements but: 1/ you must provide baseline results with a well tuned phrase based mt system
> <Response>: We have looked into this direction for comparison. However, in the IWSLT 2014 competition, test set tst2014 is not revealed. Hence, we cannot directly compare results with the all the teams on [Cettolo 2014].  Given in IWSLT 2015 English-German task, NMT outperforms Phrase-Based MT by a large margin (up to 5.2 BLEU score) [Luong 2015]. Our NPMT further outperforms the state-of-the-art NMT types of systems by up to 2.7 BLEU score in English-German task.
>
> Reference:
> [Cettolo 2014] Mauro Cettolo, Jan Niehues, Sebastian St¨uker, Luisa Bentivogli, and Marcello Federico. Report on the 11th IWSLT evaluation campaign, IWSLT 2014. In Proceedings of IWSLT, 2014.
> [Luong 2015] Minh-Thang Luong and Christopher D. Manning. Stanford Neural Machine Translation Systems for Spoken Language Domain. IWSLT’15.
>
> 5. 2/ the datasets are small ones, as well as the vocabularies, you should try with larger datasets and bpe for sake of comparison.
> <Response>: We are actively working on improving the speed of the system and exploring this approach on WMT datasets with bpe vocabularies. We plan to open source our implementations to expedite this direction.

---

### Decision · Program_Chairs · 2018-01-29
**ICLR 2018 Conference Acceptance Decision**

**Decision:**

Accept (Poster)

**Comment:**

this submission introduces soft local reordering to the recently proposed SWAN layer [Wang et al., 2017] to make it suitable for machine translation. although only in small-scale experiments, the results are convincing.